# Alternative Splicing and Isoforms: From Mechanisms to Diseases

**DOI:** 10.3390/genes13030401

**Published:** 2022-02-24

**Authors:** Qi Liu, Leiming Fang, Chengjun Wu

**Affiliations:** School of Biomedical Engineering, Dalian University of Technology, Dalian 116024, China; liuqi_dut@163.com (Q.L.); flming1018@foxmail.com (L.F.)

**Keywords:** alternative splicing, splicing factors, diseases, drugs

## Abstract

Alternative splicing of pre-mRNA is a key mechanism for increasing the complexity of proteins in humans, causing a diversity of expression of transcriptomes and proteomes in a tissue-specific manner. Alternative splicing is regulated by a variety of splicing factors. However, the changes and errors of splicing regulation caused by splicing factors are strongly related to many diseases, something which represents one of this study’s main interests. Further understanding of alternative splicing regulation mediated by cellular factors is also a prospective choice to develop specific drugs for targeting the dynamic RNA splicing process. In this review, we firstly concluded the basic principle of alternative splicing. Afterwards, we showed how splicing isoforms affect physiological activities through specific disease examples. Finally, the available treatment methods relative to adjusting splicing activities have been summarized.

## 1. Introduction

Alternative splicing is an essential process in post-transcriptional mRNA processing, and produces various mature mRNAs with different structures and functions. In this process, exons are taken together in different combinations and introns are removed. Recent data indicate that each transcript of protein-coding genes contain 11 exons and produce 5.4 mRNAs on average [1]. So far, seven basic types of alternative splicing have been identified, including exon skipping, alternative 5′-splice site, alternative 3′-splice site, mutually exclusive exons, intron retention, alternative promoter, and alternative polyadenylation [2] (Figure 1). A notable example of alternative splicing is the human gene *TTN* which encodes muscle protein titin and contains 364 coding-exons and 4039 different splicing events which have been identified by RNA-sequencing [3]. Most genes generate at least two transcript variants. The alternative spliced mRNAs are further translated into many protein variants which differ in function and structure. The precision and diversity of alternative splicing events are aided by many significant factors, such as the strength or weakness of splice sites, the concentration and combination of enhancing and silencing splicing factors, chromatin modifications, and RNA secondary architectures [3]. Since the activity of splicing factors and the spliced variants change the developing process of diseases, they can also serve as experimental indicators or biomarkers for diagnosis. 

### 1.1. Function and Assembly of Spliceosome

The spliceosome recruited by cis-acting elements and trans-acting factors plays a critical role in regulating the constitutive and alternative splicing procedure. It is an enormous macromolecular complex containing five small nuclear RNAs (U1, U2, U4, U5, and U6), and hundreds of combined proteins called small nuclear ribonucleoproteins (snRNPs). The complex splicing regulation process is carried out by virtue of dynamic assembles of snRNPs in a stepwise fashion. Here we show a brief overview. Firstly, U1 snRNP binds to 5′-ss GU di-nucleotide, splicing factor 1 (SF1) and U2AF65 bind to branch point site (BPS) and the polypyrimidine tract (PPT), respectively, forming the E complex. Secondly, U2 snRNP base-pairing interacts with the BPS displacing SF1 to form complex A. Afterwards, it recruits U4/U6/U5 tri-snRNP, leading to the formation of complex B, in which U5 snRNP binds to 3′-ss and U6 snRNP binds to U2 snRNP. Meanwhile, U1 and U4 snRNP are released. Complex C is then formed. Followed by two transesterification steps, intron is folded into a lariat and 5′-ss is cleaved. Finally, the two exons are linked together and the lariat is released. snRNPs can be used for recycling [4]. It has been indicated that mutations of splicing factors can disrupt the expression ratios of small nuclear RNAs and spliceosome assembly, inducing premature pathogenic termination of mRNA translation [5]. 

### 1.2. Essential Structures and Elements for Alternative Splicing Regulation

The boundaries between exons and introns are illustrated by a 5′-splice site (5′-ss) with highly conserved GU nucleotide combination, a 3′-splice site (3′-ss) with highly conserved AG nucleotide sequence, a branch-point sequence (BPS) located at 18–40 nucleotides upstream of the 3′-ss, and a polypyrimidine tract (PPT) which is also critical in recognizing 3′-ss. The decisions of removal or retention toward specific exons depend on the role of cis-acting elements, which are short nucleoside sequences containing RNA binding sites located on the pre-mRNA. The cis-acting elements include exonic splicing enhancers (ESEs), exonic splicing silencers (ESSs), intronic splicing enhancers (ISEs), and intronic splicing silencers (ISSs) [6]. The two major families of cellular RNA binding proteins participating in the splicing process are serine/arginine-rich (SR) protein and the heterogeneous nuclear ribonucleoproteins (hnRNPs). The structures of RNA recognition motif (RRM) and serine/arginine-rich domain (RS domain) are what make SR proteins functional in splicing regularly. For instance, SR proteins mediate the interaction between U1 snRNP and 5′-ss and recruit U2 snRNP to the 3′-ss. Besides, they often cooperate with other positive splicing factors to form enhancing complexes, such as TRA2, SRRM1, and SRRM2 [7]. However, SR proteins and hnRNPs families generally have an opposite effect during choosing the alternative splice sites and exons by definition act in a competitive manner. ESEs and ISEs mainly recruit SR proteins acting as splicing activators, while ESSs and ISSs are usually recognized by hnRNPs proteins function as splicing repressors [8]. For example, splicing of exon 6B from *β-tropomyosin* gene depends on the G-rich intronic sequence (S3) and downstream of exon 6B, which can act as either an enhancer or a silencer. ASF/SF2 and SC35 bind to S3 and positively stimulate the splicing of exon 6B, whereas hnRNP A1 competitively disrupts their interaction [9]. Mutation of spliceosome components and dysregulation of splicing mechanisms can affect the network of downstream splicing targets, causing human diseases such as cancer, neurodegenerative disorders, and metabolic diseases [10].

### 1.3. Tissue Specificity of Alternative Splicing

The distribution of alternative splicing factors is tissue-specific, which is the reason for the diversity of cell differentiation and tissue specificity. It has been indicated that more than 50% of genes express different alternative spliced isoforms among tissues. The human brain, the most functionally diverse tissue, contains several specific splicing factors, including nPTB, NOVA1, and NOVA2. In the process of neuronal differentiation, the expression of splicing factors shifted from PTB to nPTB. The upregulation of PTB is responsible for approximately a quarter of nervous system-specific alternative splicing. The CELF family of proteins containing three RRMs is broadly expressed in many tissues. Among them CELF1, CELF2, CELF5, and CELF6 are located in the brain, serving as alternative splicing regulators which mainly target gene *TNTT2*. In addition, CELF2 and CELF5 are also distributed in the heart and skeletal muscle tissues. RBM35a and RBM35b are epithelial cells-specific splicing factors, controlling the expression of epithelial characteristics related exons [7].

## 2. Alternative Splicing Factors and Spliced Isoforms in Relation to Cancer Pathogenesis

Cancer is a complex disease which can be caused by various factors. It has been determined that gene expression profiles of tumor cells are different from normal samples. Instable cellular homeostasis is an important cause of cancer, and has been reported to be closely related to aberrant alternative splicing. Since alternative splicing plays a key role in post-transcriptional regulation and controls the formation of spliced variants, the mutations and changed level of splice factors may contribute to tumorigenesis. 

Abnormal expressions of specific splicing isoforms can cause specific cancers but also can serve as a biomarkers and therapeutic targets. The TCGA database allows one to analyze the expression of alternative splicing patterns during cancer progression [11]. Increasing evidence demonstrates the dysregulation of alternative splicing leads to the production of tumor-associated isoforms that further impact cellular activity [4], such as sustaining proliferation, preventing cell death, rewiring cell metabolism, promoting angiogenesis, enabling cell invasion and metastatic dissemination, and enabling drug resistance [12]. For example, the mutation of SF3B1 is highly related with several cancers including chronic lymphocytic leukemia (CLL), cutaneous melanomas, and uveal melanomas. SF3B1 contains an essential region which interacts with SF3B14a and further forms a complex with U2AF2, playing a key role in BPS recognition. However, the mutation of SF3B1 at exons 12–15 disturbs the interaction between SF3B1 and SF3B14a, leading to the prevention of BPS recognition and 3′-ss mis-selection. Moreover, the mutated SF3B1 functions as an anti-apoptotic factor which is regularly detected in various cancers in order to sustain the cell proliferation [13]. SR proteins are responsible for multiple cellular physiological process and involved in the alteration of gene expression in tumors. SRSF1 is often upregulated in breast tumors through binding with MYC, leading to increasing of cell proliferation and decreasing of cell apoptosis. Additionally, SRSF1 overexpression in lung cancer leads to resistance to the chemotherapy drugs cisplatin and topotecan. Previous studies have demonstrated that the SRSF2 mutant is associated with myelodysplastic syndromes (MDS), since the mutated SRSF2 expression alters the binding specificity, inducing the inclusion of a premature termination codon (PTC). This PTC locates in EZH2 region which encoding a histone methyltransferase related to the pathogenesis of MDS. SRSF6 overexpression is detected in skin cancer, promoting the splicing of cassette exons and inducing hyperplasia [14]. The hnRNPs play an important role in regulation of pre-mRNA splicing. Abnormal expression of hnRNPs affects RNA splicing, results in alteration of RNA expression levels, and further causes the occurrence of cancer. As the most abundant hnRNPs, hnRNP A1/A2 has been reported to be upregulated in lung cancer. Overexpression of hnRNP A1/A2 may function as carcinogenic factor to promote cell proliferation. Moreover, hnRNP A1 and hnRNP A2 participate in recognizing and protecting telomeric sequences. Thus, they are related to cancer regulation. In contrary, silencing of hnRNPA1/A2 expression causes tumor cell apoptosis [15]. In addition, miRNAs can modulate splicing-factor expression and function as oncogenes. miR-30a-5p and miR-181a-5p regulate SRSF7 in renal tumors, thereby altering the splicing pattern [16,17]. Overexpression of splicing associated miRNAs has been detected in a variety of cancers. For example, repression of SRSF1 results in upregulation of miR-10a and miR-10b, thereby promoting terminal differentiation of neuroblastoma cells.

Alternative splicing events generate protein isoforms related to cancer hallmarks, promoting tumorigenesis. Gene *RPSkKB1* alternatively encode two protein isoforms, full-length RPS6KB1-1 and RPS6KB1-2 lacking kinase domain. The production of RPS6KB1-2 is regulated by SRSF1 mediated AS, and contributes to tumor growth in lung and breast cancer, while RPS6KB1-1 suppresses proliferation of cancer cells [15,18]. Similarly, oncogene *CCND1* encoding cyclin D1 has two isoforms, namely conventional cyclin D1a and cyclin D1b lacking the C-terminal protein domains. Cyclin D1b C-terminal domain encoded by the exon 5 is a GSK-3β phosphorylation site, allowing the cyclin D1b to be transported from the nucleus to the cytoplasm. Altered selection of 5′-ss induce exclusion of exon 5, thereby causing cyclin D1b to be trapped in the nucleus. As opposed to cyclin D1a, the overexpression of cyclin D1b variant was observed in breast cancer tissues compared to normal breast tissues, which characterizes metastasis and invasive migration of cancer cells mediated by αvβ3 and toll-like receptor 4 (TLR4) [19,20,21,22,23] (Figure 2). 

## 3. The Relationship between Alternative Splicing and Neurological Diseases

Neurodegenerative disease constitutes a variety of mental and neuromuscular disorders, including Alzheimer’s disease (AD), Parkinson’s disease (PD), spinal muscular atrophy (SMA), and familial dysautonomia (FD). It has been found that alternative splicing plays an important role in neurological diseases.

Neuro-oncological ventral antigens 1 and 2 (NOVA1 and NOVA2) are two members of the NOVA family, regulating neuron-specific alternative splicing. They belong to splicing factors and mainly enriched in brain tissue and are responsible for neuronal viability and maturity. NOVA1 is distributed in the cerebellum and spine, whereas NOVA2 is located in the cortex. NOVAs regulates various receptors and voltage-gated channels which are critical for the signal transduction of neuromuscular junctions. As an alternative splicing method for a key regulator of glycine α2 exon 3A (GlyRα2E3A) pre-mRNA splicing, NOVA1 recognizes an adjacent intronic splice site, alternatively inducing exon 3A inclusion. NOVA1, together with TDP-43, participates in RNA recognition and induces various neurodegenerative diseases such as schizophrenia and amyotrophic lateral sclerosis [24]. NOVA1 participates in the splicing of RNA-binding protein RBM8A and regulates RNA metabolism, causing neurological damage-like symptoms of AD. 

Alternative splicing events frequently occur in the progression of AD. FynT is an isoform of tyrosine kinase protein Fyn, which is upregulated in AD patients and correlated with chronic neuro-inflammation [25]. Alternative splicing of exon 15 of Amyloid precursor protein (APP) transcript changes the ratio of APP proteolytic fragments betaA4 and p3, which contributes to the formation of amyloid plaque and aggravate AD [26]. Another major toxic in AD is microtubule-stabilizing protein tau. Tau has two isoforms, four microtubule repeats (4R-tau), and three microtubule repeats (3R-tau). The generation of both isoforms are determined by the inclusion or skipping of exon 10, which are further regulated by splicing factor SRSF6 [27]. The two isoforms expression level of 3R-tau and 4R-tau have to maintain a balance in healthy individuals. However, dysregulation of tau splicing disturbs the ratios, thereby causing neurofibrillary degeneration in AD patients [28]. AMPA serves as a critical mediator of synaptic transmission. AMPA receptor subunits of PD patients have been regulated alternatively, which are potential biomarkers for PD diagnosis and therapies [29]. SMA is a common autosomal recessive disease which is partially caused by the decrease of SMN protein. Human beings encoded two *SMN* genes, namely *SMN1* and *SMN2*. They are almost identical and encode SMN protein, which is crucial for snRNP assembly. However, *SMN2* is vulnerable to mutate in exon 7 due to the replacement of C by T at position 6. Hence, during the post transcriptional processing, exon 7 is skipped in *SMN2* pre-mRNA splicing, generating truncated isoforms which cannot compensate the function of SMN [30,31]. Sodium butyrate has been detected to modulate exon 7 inclusion of *SMN2* transcript, thereby producing the full-length SMN protein [32]. Familial dysautonomia (FD) is a recessive disease mainly caused by the mutation of the i-kappa-B kinase complex associated protein (IKBKAP). The mutated IKBKAP loses its function, and thereby FD patients show a demyelination phenotype. Single-base changes in less well-conserved nucleotides affecting the selection of splice site are responsible for numerous genetic diseases. In this case, the mutation of nucleotide T to C occurring at position 6 of intron 20 disrupts the interaction between IKBKAP pre-mRNA and U1 snRNA, and weakens the splicing from the intronic 5′-ss. Additionally, computational analyses identified that upstream 3′-ss also contains weak splicing signals, further causing the defining of exon 20 almost impossible. [33] (Figure 3).

## 4. Alternative Splicing in Metabolic Diseases: Diabetes and Obesity

Diabetes mellitus (DM) is the most common metabolic disease, with a high incidence rate worldwide. About 422 million diabetes patients have been diagnosed worldwide. Type 1 DM is caused by excessive apoptotic of pancreatic β cells which secrete insulin, leading to the disruption of glucose uptake. Type 2 DM is induced by insulin resistance due to the lower activity of insulin receptor (IR) in the liver, skeletal muscle, and adipose tissues. The gene *INSR*, located on chromosome 19 and containing a total of 22 exons, encodes two isoforms IR-A and IR-B through alternative splicing events [34]. IR-B includes exon 11 during pre-mRNA splicing, whereas IR-A excludes it (Figure 4). Whether exon 11 is included depends on the recognition and activation of the exonic splicing enhancer (ESE) and exonic splicing silencer (ESS) on it. The ESE in exon 11 has been detected to bind to SRp20 and SF2/ASF, the inclusion of exon 11 is blocked when knockdown SF2/ASF, whereas, the inclusion of IR exon 11 is repressed by CUG-BP1 through interacting with ESS [35]. In terms of function, IR-B increases β cell survival by affecting the MAPK/ERK signaling pathway. Hence, it improves the expression level of SRSF1. It has been reported that some splicing factors are involved in the splicing of IR-B receptor, such as hnRNPA1, SRSF7 [36], SRSF3 [35], and NOVA1 [37]. In addition, the disruption of IR-A/IR-B ratio caused by alternative splicing can trigger diabetes and complications [38].

Obesity seriously affects human life and health and increases the risk of diabetes and hypertension. Animal models indicate that obese mice have hypercholesterolemia and hyperinsulinemia. The uptake ability of leptin is closely related to leptin receptor (obesity receptor, Ob-R). The Ob-R expression is regulated by alternative mRNA splicing that producing many spliced isoforms, including Ob-Rb, Ob-Ra, Ob-Rc, Ob-Rd, and Ob-Re. The full-length isoform Ob-Rb is responsible for maintaining energy balance, and is mainly expressed in immunocyte to activate the immune response [39]. The deletion of soluble isoform Ob-Re can lead to severe obesity in mice as well as to an increase BMI in humans [39,40]. Alternative splicing event generates three short leptin isoforms (Ob-Ra, Ob-Rc, and Ob-Rd) that lack of the full length exon 19. They are commonly responsible for internalization and degradation of leptin. Increasing of Ob-Ra in db/db mice results in their early obesity phenotype. The Ob-Re soluble isoform participates in the regulation of serum leptin concentration. Lipin-1, which is encoded by *LPIN1* and is a key transcriptional coactivator required for lipid metabolism and adipocyte differentiation, expresses two protein isoforms (lipin-1A and lipin-1B). Lipin-1A is responsible for the early differentiation of adipocytes. Inclusion of exon7 in *LPIN1* results in lipin-1B, whereas skipping of exon 7 generates lipin-1A. Experiments conducted on SFRS10 heterozygous mice showed that SFRS10 targets *LPIN1*, suggesting that reduced SFRS10 results in higher expression of lipin-1B [41]. Functionally, lipin-1A plays a key role in the early stage of adipocyte differentiation, directing the expression of adipogenic transcription factors PPARγetc. Iipin-1B predominates in mature adipocytes, inducing the expression of lipogenic genes.

## 5. Therapeutic Based on Modulating Alternative Splicing

With the pathogenesis of different diseases having been elucidated, numerous studies have demonstrated that pre-mRNA processing (and especially pre-mRNA splicing) plays a vital role in disease occurrence and progression. Since alternative splicing events are tissue-specific or disease-specific, scientists have attempted to take advantage of alternative splicing mechanisms to develop biomarkers or drugs targeting correlative diseases. Currently, novel drug designs based on alternative splicing mechanisms are rapidly being developed. Some therapies targeting elements of pre-RNA splicing are being researched (e.g., RNAi-based isoform specific modulation) [42].

RNA splicing modulation can be achieved by affecting the activity of spliceosome components to impact splicing efficiency or by directly targeting the aberrant isoforms. FR901464 is a natural compound, acting as splicing inhibitors in clinical treatment. It has been clarified that the interaction between FR901464 and SF3B1 inhibits pre-mRNA splicing in HeLa cells, thereby preventing proliferation. However, nearly 90% of splicing events regulated by SF3B1 cannot be inhibited [43,44], suggesting that a strong splice site is less sensitive to this treatment. Similarly, the synthetic analog of FR901464, including Sudemycin E and Meayamycin, lowered splicing capacity by associating with U2 snRNP component SF3B1 [43,45]. Another splicing inhibitor is Isoginkgetin (7-O-β-D-glucopyranoside), which can prevent recruitment of the U4/U6-U5 tri-snRNPs and block the formation of spliceosome complex B [46]. In addition, a class of synthetic derivatives including E7107, spliceostatin A (SSA), meayamycin B (MAMB), sudemycins, and H3B-8800 serve as SF3B1 inhibitors with high specificity [47]. These small molecules disrupt the formation of spliceosome complexes by binding to SF3B1, affecting the function of spliceosomes and exon ligation throughout the catalytic cycle [48,49]. Further, targeting protein kinases that regulate splicing factors can also change splicing patterns, such as CDC2-like kinase (CLK) families [50]. For instance, CLK inhibitor SM09419 exhibits strong anti-proliferative ability in acute myeloid leukemia (AML) cells by inhibiting Wnt pathway [51]. MDM4, an upstream negative regulator of tumor suppressor p53, is usually up-regulated in cancer patients, resulting in apoptosis inhibition and tumor growth [52]. Studies have shown that the MDM4 has eight alternative spliced isoforms, in which full length MDM4 (MDM4-FL) contains all domains of MDM4 and has an inhibitory function for p53. MDM4-S lacks exon 6, which is considered to be a non-sense mediated decay (NMD) switch. MDM4-S is very unstable, so the transition from MDM4-FL to MDM4-S can shut down the expression of MDM4-FL, thereby activating p53. Splicing factors involving in MDM4 exon 6 inclusion include SRSF3 and PRMT5 [53,54]. Pladienolide B and GSK3203591 serve as their inhibitors, respectively, leading to p53 up-regulate [55,56]. Since full knowledge of these small molecules s lacking, off-target effects sometimes occur, which limits their clinical value. Therefore, the exact mechanisms underlying these compounds remains to be investigated. 

Another approach approved by the FDA is antisense oligonucleotide (ASO), which is based on technology used to modify RNA splicing with complementary sequences. Antisense oligonucleotides (ASOs) are short oligonucleotides containing 15–25 bases, which can switch pre-mRNAs splicing sites in order to prevent the production of protein isoforms causing diseases. Splice switching oligonucleotides (SSO) can target specific cis-elements within the pre-mRNA and compete with splicing factors, determining whether exons and introns are retained or not [42]. Spinraza™ was the first treatment adopting ASO method aimed at SMA [57]. Splice-switch ASO is used in tumor therapy to induce the apoptosis of cancer cells. For example, in breast cancer, the gene *STAT3* has two expression isoforms, namely STAT3α and STAT3β. By targeting splicing enhancer and regulating the inclusion of exon 23, ASO-mediated treatment increases the expression of the STAT3β isoform (Figure 5), which lacks the C-terminal transactivation domain, causing cell-cycle arrest and apoptosis of cancer cells. In xenograft breast cancer models, upregulation of STAT3β causes tumor regression. Effective MDM4, ERBB4, BRCA2, and GLDC pre-mRNAs splicing regulation based on ASO technology has also been developed.

## 6. Conclusions

This review describes that alternative splicing is a complex physiological process integrating many regulatory factors. Correct alternative splicing depends on the appropriate cis-acting elements, every accurate step of spliceosome assembly, specific RNA-protein interactions, and numerous splicing factors in the tissue-specific manner. A large number of previous articles and ongoing studies on alternative splicing have elucidated how the interaction between transcriptomes and proteomes affects the cellular environment and leads to diseases, shedding light on promising precision medical technology. In cancer patients, abnormal splicing factors fail to manipulate the distinguish of exon-intron junctions and cause pathogenic spliced isoforms, inducing proliferative, invasive, and migratory phenotype of tumors. Some neurological diseases can be attributed to false definition of exon-intron junctions caused by point mutations. However, due to the complexity of brain tissue and nervous system, the related mechanisms still need to be further studied. We demonstrated some alternatively spliced protein variants playing role in metabolic diseases, which provide novel treatment targets to compensate splicing dysregulation. Furthermore, more and more drugs designed to target modulating splicing patterns have been developed, which has provided hope for prolonging the lifespan of human beings. Nevertheless, our understanding of the regulatory network of splicing factors is still insufficient. Genome-wide alternative splicing detection and prediction technologies should be applied to expand our knowledge on alternative splicing mediated regulatory mechanisms. 

## Figures and Tables

**Figure 1 genes-13-00401-f001:**
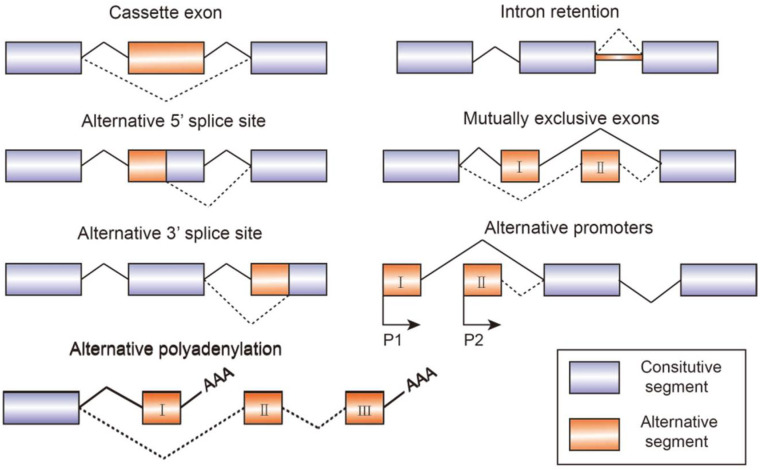
Schematic figure representing seven modes of alternative splicing of pre-mRNA: cassette exon, intron retention, alternative 5′ splice site, mutually exclusive exons, alternative 3′ splice site, alternative promoters, and alternative polyadenylation. Alternative splicing sites are connected by dashed lines. Boxes represent exons and lines represent introns.

**Figure 2 genes-13-00401-f002:**
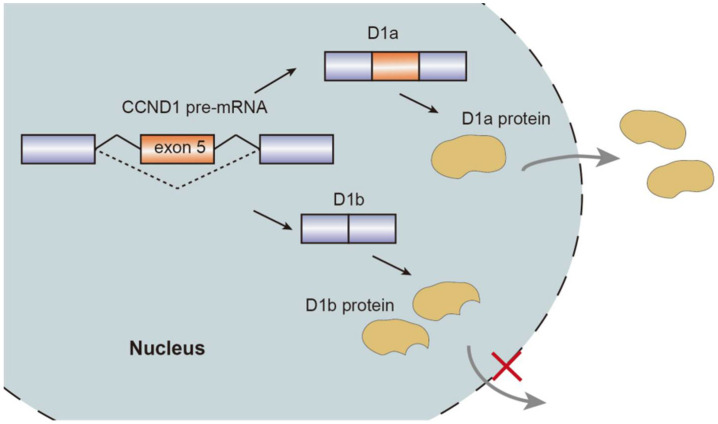
Schematic representation of CCND1 pre-mRNA splicing mechanism, isoforms, and nuclear transport.

**Figure 3 genes-13-00401-f003:**
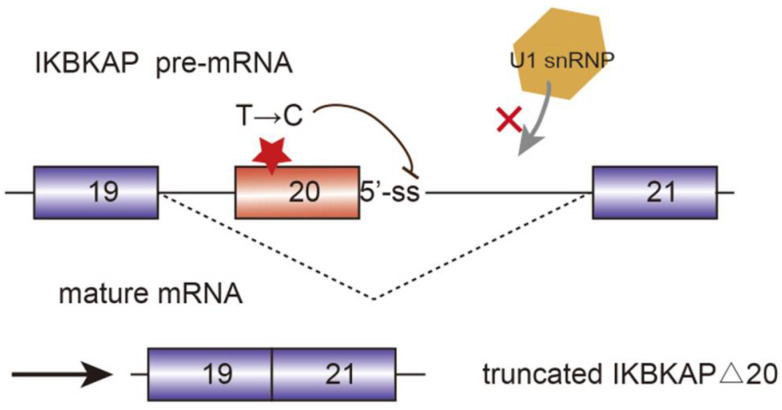
Mutated i-kappa-B kinase complex associated protein (IKBKAP) plays a role in the development of familial dysautonomia (FD). The single nucleotide mutation from T to C disrupts the interaction between pre-mRNA and U1 snRNP and induce the exon 20 exclusion.

**Figure 4 genes-13-00401-f004:**
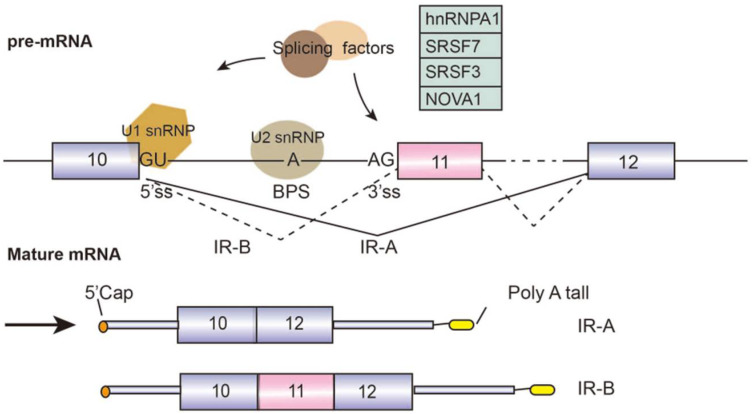
Graphical representation of assembly of alternative splicing factors (hnRNPA1, SRSF7, SRSF3, and NOVA1) on the *INSR* pre-mRNA and insulin receptor (IR) splice variants. Splicing factors are listed in the green box. Alternative splicing sites are connected by dashed lines. The boxes represent exons and the lines represent introns.

**Figure 5 genes-13-00401-f005:**
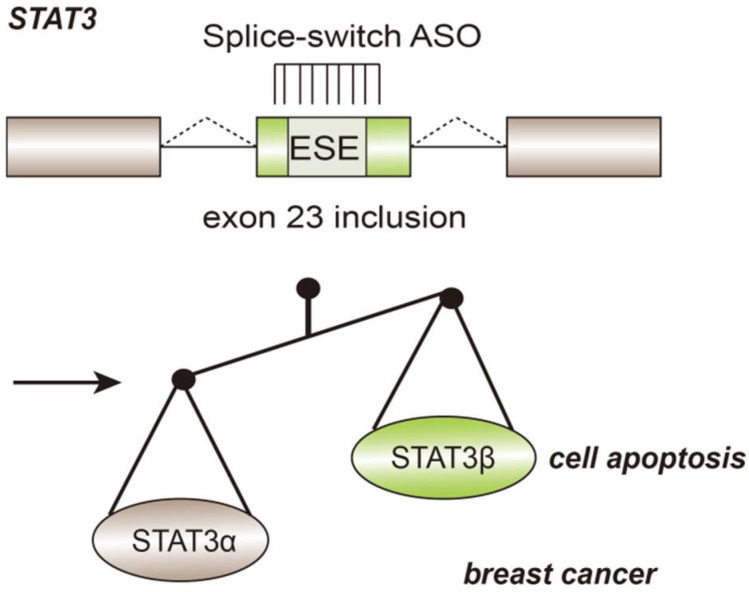
Application of splice-switch ASO in the treatment of breast cancer. Oncogene *STAT* expresses two isoforms: STAT3α and STAT3β. ASO targets the ESE and induce exon23 inclusion, thereby improving the production of STAT3β and leading to cell apoptosis in breast tumors.

## Data Availability

Not applicable.

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
