# Peer review of "Alternative Splicing and Isoforms: From Mechanisms to Diseases"

_genes, 2022, doi:10.3390/genes13030401_

Round 1

Reviewer 1 Report

The review work presented by Qi Liu and colleagues entitled “Alternative splicing and isoforms: from mechanisms to diseases” is well written, clear, and easy to read. The topic is interesting and therefore, it adds clustered information to the area of RNAs/nucleic acid biology. The part linked to microRNAs controlling splicing protein factors is a cutting edge area and I suggest to consider in future review work.

Minor

Please check the misspelling along with the text. Figure 1 took off “This” and start the sentence with "Schematic"

Either no to used abbreviation, leave “alternative splicing” and took off “(AS)”

Author Response

Response to Reviewer 1 Comments

Point 1: Please check the misspelling along with the text. Figure 1 took off “This” and start the sentence with "Schematic"

Response 1: I have checked the spelling throughly and correct the mistakes. In figure 1 legend, "This" has been taken off.

Point 2: Either no to used abbreviation, leave “alternative splicing” and took off “(AS)”

Response 2: Thank you for your advice, I have abandoned using abbreviation, and use "alternative splicing" directly instead of "AS".

Reviewer 2 Report

The authors aim to present the mechanism of alternative splicing and its relevance in different diseases. The topic selection is important, as it really represents a major molecular mechanism.

Despite the title, the mechanisms of alternative splicing are only very superficially presented. The molecular mechanisms leading to alterations of alternative splicing, are very briefly presented. It would be rather advantageous to expand the discussion of these parts and to present the mechanisms in more detail.

The later parts of the manuscript, disease groups are presented with some selected examples of alternative-splicing-mediated pathomechanisms. The examples seem somewhat arbitrary, and a general concept of mechanism is lacking.

The English language needs major revision. 

Author Response

Response to Reviewer 2 Comments

Point 1: Despite the title, the mechanisms of alternative splicing are only very superficially presented. The molecular mechanisms leading to alterations of alternative splicing, are very briefly presented. It would be rather advantageous to expand the discussion of these parts and to present the mechanisms in more detail.

Response 1: Thank you for your advice. In the “introduction” section of the article, I added a new paragraph to elaborate the molecular mechanism of alternative splicing process, and highlighted the significance and complexity of alternative splicing.

Point 2: The later parts of the manuscript, disease groups are presented with some selected examples of alternative-splicing-mediated pathomechanisms. The examples seem somewhat arbitrary, and a general concept of mechanism is lacking.

Response 2: We have taken your suggestion into serious consideration. We selected the typical example from each disease group, and enrich the details of molecular mechanism, making the example more credibility.

Point 3: The English language needs major revision. 

Response 3: We have edited the language of the article and improved the grammar.

Round 2

Reviewer 2 Report

The manuscript still lacks a general, comprehensive presentation of mechanisms.

Round 3

Reviewer 2 Report

My proposals have been performed now. The manuscript has been significantly improved and I thus propose its acceptance.